# Phytochemical Analysis and Antioxidant and Anti-Inflammatory Capacity of the Extracts of Fruits of the *Sechium* Hybrid

**DOI:** 10.3390/molecules25204637

**Published:** 2020-10-12

**Authors:** Itzen Aguiñiga-Sánchez, Marcos Soto-Hernández, Jorge Cadena-Iñiguez, Mario Suwalsky, José R. Colina, Ivan Castillo, Juana Rosado-Pérez, Víctor M. Mendoza-Núñez, Edelmiro Santiago-Osorio

**Affiliations:** 1Hematopoiesis and Leukemia Laboratory, Research Unit on Cell Differentiation and Cancer, FES Zaragoza, National Autonomous University of Mexico, 09230 Mexico City, Mexico; liberitzen@yahoo.com.mx; 2Interdisciplinary Research Group of *Sechium edule* in Mexico (GISeM), Texcoco, Agustín Melgar 10 Street, 56153 Texcoco, Mexico; msoto@colpos.mx (M.S.-H.); jocadena@gmail.com (J.C.-I.); 3Postgraduate College, Campus Montecillo, Km 36.5 Mexico-Texcoco Highway, 56230 Texcoco, Mexico; 4Postgraduate College, Campus San Luis Potosí, Iturbide No. 73 Street, Salinas de Hidalgo, 78600 San Luis Potosí, Mexico; 5Facultad de Medicina, Universidad Católica de la Santísima Concepción, 4090541 Concepción, Chile; msuwalsk@udec.cl; 6Facultad de Ciencias Químicas, Universidad de Concepción, 4070386 Concepción, Chile; jcolina@udec.cl; 7Instituto de Química, Universidad Nacional Autónoma de México, Circuito Exterior, Ciudad Universitaria, 04510 Mexico City, Mexico; joseivan@unam.mx; 8Research Unit on Gerontology, FES Zaragoza, National Autonomous University of Mexico, 09230 Mexico City, Mexico; juanaropez@yahoo.com.mx

**Keywords:** chayote, phenolic compounds, glutathione peroxidase, TNFα, IL-10

## Abstract

In addition to their own antioxidants, human cells feed on external antioxidants, such as the phenolic compounds of fruits and vegetables, which work together to keep oxidative stress in check. *Sechium edule*, an edible species of chayote, has phenolic compounds with antioxidant activity and antineoplastic activity. A *Sechium* hybrid shows one thousand times greater antineoplastic activity than edible species, but its antioxidant and anti-inflammatory activities and the content of phenolic compounds are unknown. The aim of this study was to determine the antioxidant and anti-inflammatory capacity of the extract of fruits of the *Sechium* hybrid in vitro and in vivo. Phytochemical analysis using HPLC showed that the extract of the *Sechium* hybrid has at least 16 phenolic compounds; galangin, naringenin, phloretin and chlorogenic acid are the most abundant. In an in vitro assay, this extract inhibited 2,2-diphenyl-L-picrylhydrazyl (DPPH) activity and protected the dimyristoylphosphatidylethanolamine (DMPE) phospholipid model cell membrane from oxidation mediated by hypochlorous acid (HClO). In vivo, it was identified that the most abundant metabolites in the extract enter the bloodstream of the treated mice. On the other hand, the extract reduces the levels of tumor necrosis factor alpha (TNFα), interferon gamma (IFNγ), and interleukin-6 (IL-6) but increases interleukin-10 (IL-10) and glutathione peroxidase levels. Our findings indicate that intake of the fruits of the *Sechium* hybrid leads to antioxidant and anti-inflammatory effects in a mouse model. Therefore, these results support the possibility of exploring the clinical effect of this hybrid in humans.

## 1. Introduction

The biological activity of the cell implies an exquisite control of cellular metabolism, including the respiratory chain, in which oxidizing molecules are inevitably generated, a process during which an oxygen molecule loses an electron and a superoxide anion is generated; under normal conditions, the latter is rapidly converted into hydrogen peroxide by the enzyme superoxide dismutase and then converted into water by catalase and glutathione peroxidase, thus avoiding free radical generation and oxidative stress damage and ensuring the viability and cellular functionality of any tissue [1]. Excessive production of prooxidants and a shortage of antioxidants leads to oxidative stress that compromises cell viability and body health [2].

The inability of the antioxidant system to compensate for the excess of prooxidant agents leads to destabilization and dysfunctionality of the cells and therefore of the tissue of which they are constituents due to the exacerbation of prooxidant and proinflammatory activity, which can promote the development of chronic noncommunicable diseases (NCDs), such as type 2 diabetes mellitus, stroke, heart attack, cancer or chronic liver diseases [1]. These diseases have a strong impact on public health since they constitute the top five causes of death in the world despite the availability of conventional (allopathic) therapies to treat each of these diseases [3]. Human cells, in addition to their own antioxidants, are nourished by external antioxidants, particularly the phytochemicals of fruits and vegetables, such as phenolic compounds including flavonoids, phenolic acids, anthocyanins, and flavonones, among others, which work together with endogenous antioxidants to prevent oxidative stress [4].

*Sechium edule* (Jacq.) Sw. is an edible plant species of Mesoamerican origin whose fruit extract contains nonphenolic alkaloids, saponins, sterols, triterpenoids, flavonoids, and phenolic acids [5], as well as glycosylated flavonoids [6], with antihypertensive [7], antimicrobial [8], antioxidant [9], antitumor [10], nephroprotective [11], anti-inflammatory and hepatoprotective properties [12].

Recently, it has been shown that the extract of the edible *Sechium edule* var. *nigrum spinosum* inhibits tumor cell line proliferation without damaging normal cells [5] and has antioxidant properties [13] that involve several phenolic compounds, including naringenin [5], a recognized antioxidant agent [14]. On the other hand, the Mexican Interdisciplinary Research Group on *Sechium edule* (GISeM in Spanish) developed a *Sechium* hybrid that was generated from the *Sechium edule* varietal group wild type II chayote, resulting in the *Sechium* H387 07 hybrid. This hybrid was registered in the Mexican National Catalog of Plant Varieties (CNVV in Spanish), which is managed by the Mexican National Seed Inspection and Certification Service (SNICS in Spanish) [15]. The extract of the fruit of *Sechium* H387 07 was identified as being one thousand times more active in inhibiting tumor cells than that of its edible relative *Sechium edule* var. *nigrum spinosum*, with none of the varieties damaging normal cells [16]. The relationship of the antineoplastic, antioxidant and anti-inflammatory activity of some natural compounds is known [17]; thus, the extract of the fruit of *Sechium* H387 07 has antineoplastic activity, but it is unknown whether it possesses antioxidant and anti-inflammatory activities and phenolic compound contents. Therefore, the objective of this study was to analyze the content of phenolic compounds and the antioxidant and anti-inflammatory activity of the extract of fruits of the *Sechium* H387 07 hybrid.

## 2. Results and Discussion

### 2.1. Phytochemical Analysis of the Extract of Fruits of the Hybrid of Sechium H387 07

The total phenolic compound content of the extract of fruits of the *Sechium* H387 07 hybrid was 36.18 mg of gallic acid equivalents (GAE) per gram of extract. This represents 3.618% of the total phenolic compounds present in the extract.

Once the presence of phenolic compounds in the extract was identified, the presence of flavonoids was analyzed by HPLC. Of the eight flavonoid compounds used as standards, the presence of rutin, phlorizin, myricetin, quercetin, naringenin, phloretin, galangin and apigenin was detected (Figure 1A,B), of which the highest to lowest concentrations were galangin, phloretin, naringenin and rutin, with 21.94, 4.616, 3.304 and 1.273 mg/g extract, respectively, while the rest of the flavonoids showed concentrations of less than 1 mg/g extract (Table 1). Of the eight phenolic acid compounds used as standards, gallic acid, chlorogenic acid, syringic acid, vanillic acid, *p*-hydroxybenzoic acid, caffeic acid, ferulic acid and *p*-coumaric acid were identified (Figure 1C), and the chemical structures of the flavonoids and phenolic acids in the extract of fruits of hybrid H387 07 were drawn (Figure 2). On the other hand, the highest presence was chlorogenic acid at 4.224 mg/g of extract, followed by caffeic acid at 0.187 mg/g of extract, while the rest of the phenolic acids were identified in trace amounts (Table 1). In addition to the 16 phenolic compounds used as controls, the hybrid contains at least twice as many peaks of compounds that we could not identify due to the lack of corresponding standards. In fact, based on the retention time from 15 to 17.5 min at 254.4 nm (Figure 1A) or 2.5 to 5 min at 316.16 nm (Figure 1B) for flavonoid compounds and 1.8 to 2.7 min (Figure 1C) for phenolic acids, the hybrid contains high concentrations of unidentified compounds that are not present in its relative (5), so it would be interesting in the future to use a more sensitive method for identification, such as HPLC/mass spectrometry.

Here, it is shown that the hybrid contains 16 polyphenols, of which eight are flavonoids and eight are phenolic acids (Table 1), while one of the parents, *Sechium edule* var. *nigrum spinosum*, only contains four flavonoids and six phenolic acids [5]. The hybrid contains rutin, myricetin, quercetin and galangin, or syringic acid and ferulic acid, which are all absent in its parent; on the other hand, galangin, phloretin and naringenin were the most increased in concentration in the hybrid regarding *S. edule* var. *nigrum spinosum*; in fact, galangin increased from zero [5] to 22 mg/g of extract in the hybrid H387 07. Among the phenolic acids, chlorogenic acid underwent the largest increase, from 0.823 [5] to 4.224 mg/g of extract in the hybrid, and was the most abundant in both the parent and the hybrid, which is not surprising since the same has been observed in extracts of other plants, such as *Mahonia aquifolium* (Pursh) Nutt [18]. These data show a higher content and richness of phytochemicals in hybrid H387 compared to one of its progenitors.

### 2.2. In Vitro Antioxidant Activity of Sechium H387 07 Fruit Extract

The phytochemicals found in the fruit extract of the *Sechium* H387 07 hybrid have been reported from other plants to be antioxidants [19,20]. Due to its ability to act as a free radical, the 2,2-diphenyl-L-picrylhydrazyl (DPPH) test is used as a routine in vitro assay for the evaluation of the antioxidant properties of different important natural compounds in medicine, food and cosmetics [21]. In this sense, the potential of the extract of the *Sechium* H387 07 hybrid to reduce free radicals was analyzed, and it was found that, with a dose of 0.5 mg/mL, the inhibition of DPPH activity was reduced by almost 50%. The average dose of calculated inhibition (IC_50_) was 0.88 ± 0.018 mg/mL, and at a dose of 1.5 mg/mL of extract, approximately 70% inhibition was obtained (Figure 3). In fact, a high correlation was found between antioxidant activity and content of phenolic compounds, with a Pearson correlation of 0.995; therefore, it is undeniable that the hybrid extract has antioxidant activity in vitro, as has been reported for other similar extracts from other plants [18].

### 2.3. Protection of Model Membrane Structure by Sechium H387 07 Hybrid Extract

Another alternative to measure the antioxidant effect in vitro is based on molecular models of phospholipid membranes such as dimyristoylphosphatidylcholine (DMPC) and dimyristoylphosphatidylethanolamine (DMPE), which are located in the outer and inner monolayers of the erythrocyte cell membrane, respectively [22]. This study model was used to corroborate the antioxidant effects of chayote extract, together with hypochlorous acid (HClO), a potent oxidizing agent that damages bacteria, endothelial cells, tumor cells and red blood cells [23]. The results indicate that HClO, in a concentration-dependent manner, alters the morphology of DMPE phospholipids but not DMPC (Figure 4A,B). Therefore, it was decided that we would use a dose of 10 ng/mL HClO to assess the effect of antioxidant extract on the morphology of the phospholipid DMPE. The addition of 10 ng/mL HClO damaged the DMPE, but the addition of *Sechium* H387 07 hybrid extract to the system protected it from a concentration of 3.6 µg (Figure 4C). Hypochlorous acid is an oxidizing biological product generated by neutrophils and monocytes that functions as a potent antimicrobial agent. It is considered one of the important factors that cause tissue damage during inflammation. HClO reacts with a wide range of biological molecules that include lipids, proteins and nucleic acids. It also compromises the function of important membrane proteins, which causes alterations in the elasticity of the erythrocyte membrane. HClO induces changes in the fluidity, surface area and inhibition of Na^+^, K^+^ and Mg^2+^ flux in the membrane, as well as affecting ATPase activities, the oxidation of thiol groups and the formation of membrane chloramine, and even causes cellular lysis [23,24]. Therefore, the ability to protect the structure of the membrane phospholipid DMPE from oxidation by HClO indicates an indisputable antioxidant ability of the hybrid H387 07 extract.

### 2.4. Phenolic Compounds from Sechium H387 07 Fruit Extract that Enter the Bloodstream

There are thousands of different phenolic compounds found in plant-based foods, and it is estimated that humans’ daily intake is between 7 and 70 mg, but this amount depends largely on diet, habits and country, and these compounds represent an important part of the daily intake of bioactive compounds [25]. Flavonoids such as naringenin, quercetin, hesperetin, and glycinein and phenolic acids such as chlorogenic acid, gallic acid and caffeic acid can be identified in plasma after ingestion [26,27,28,29,30]. To assess whether the administration of the extract in the peritoneal cavity of treated healthy mice has the ability to cross into the bloodstream, the most abundant phenolic compounds in the extract were evaluated. Galangin was the main flavonoid identified in the serum of treated and untreated mice and, at baseline in the vehicle-treated group, there was an approximate concentration of 3000 ng/mL of serum, which is understandable since this metabolite is one of the components of a food supplement for rodents as it is present in the sugarcane *Saccharum officinarum* L. [31]. The 8 mg/kg hybrid extract increased the concentration of galangin in the bloodstream from the first hour after treatment to concentrations of more than 7000 ng/mL, as did the concentrations of 125 and 250 mg/kg, all of which were increased by more than 1000 ng/mL of serum with respect to the control, even after 24 h of receiving the treatment (Figure 5A). 

On the other hand, the presence of serum phloretin and naringenin was not detected in control mice, but the administration of 8 mg/mL extract showed the highest amount of 6658 ng/mL of phloretin at six hours after treatment (Figure 5B), while naringenin, at a dose of 125 mg/kg extract, increased to 2274 ng/mL at six hours and remained at 1000 ng/mL at 24 h (Figure 5C). These data are not surprising since phloretin (phloretin-2/beta-glucose) corresponds to a group of flavonoids known as dihydrochalcones that are considered antibiotic agents [32] and in a study in rats in which phloretin and phlorizine were administered only in food, the presence of plasma phloretin was detected in the first minutes, while phlorizine had a more delayed effect because phlorizine is not easily absorbed [33]. On the other hand, the data on naringenin correlate with previous studies because rats that only received naringenin in a single administration had detectable levels in the plasma at three hours post treatment, reaching their maximum peak at six hours and maintaining a presence even at low concentrations for up to 24 h [34]. In another study, healthy volunteers were given an orange juice that contained a high content of naringenin, and a maximum peak was observed at 4 h post intake, reaching concentrations of up to 3000 ng/mL [35]. Our data confirm that when treating mice with the extract of the *Sechium* H387 07 hybrid, the presence of phloretin and naringenin was identified even up to 24 h after treatment, reaching levels close to 1500 and more than 500 ng/mL, respectively, with all treatment doses.

Chlorogenic acid, the most abundant of the phenolic acids, was not found in the control mice during the first 24 h, but in mice given a dose of 250 mg/kg extract chlorogenic acid levels reached more than 11,000 ng/mL at 12 h after treatment and this was maintained until 24 h (Figure 5D).

### 2.5. Reduction in Inflammatory Cytokines in Healthy Mice Treated with Sechium H387 07 Fruit Extract

Naringenin has been described as an antioxidant agent at the extracellular and cellular levels, particularly at the intramembrane, cytoplasmic and nuclear levels, and even has epigenetic activity [1]. Here, it is shown that the administration of extract intraperitoneally to mice after 3, 6, 12 and 24 h reveals the presence of naringenin, galangin, phloretin, and serum chlorogenic acid, which indicates that these compounds have the potential to exercise systemic activity, as has already been published for the extract of fruits of *Sechium edule* var. *nigrum spinosum* [5]. To detect a systemic biological effect induced by the extract, the sera of healthy mice treated with or without the extract of the *Sechium* hybrid were analyzed for changes in the levels of the inflammatory and anti-inflammatory cytokines TNFα, INFγ, IL-6 and IL-10. The results indicate that the extract reduces the production of inflammatory cytokines such as TNFα, INFγ and IL-6 at doses of 250, 500 and 1000 mg/kg, while the dose of 8 mg/kg only reduces the production of TNFα and IL-6. In contrast, IL-10, an anti-inflammatory cytokine, was increased at all doses of the extract (Table 2). These data indicate that the extract contains metabolites that reduce the levels of proinflammatory cytokines but increase those with anti-inflammatory activity.

### 2.6. Sechium H387 07 Hybrid Extract Reduces Oxidative Stress Damage in Mice

Having shown that the extract modulates the production of proinflammatory cytokines and taking into account that antioxidant and anti-inflammatory activity are shared by several compounds [18], the antioxidant activity of the extract was evaluated in an in vivo model. The glutathione peroxidase (GPx) test has high stability in multiple tests and high levels of accuracy, and it is a widely reproducible parameter for measuring antioxidant activity [36]. It was used to evaluate the effect of the extract on the levels of the GPx enzyme in erythrocytes. The results indicate that the erythrocytes of mice without any treatment have high levels of GPx, but when damage is induced with carbon tetrachloride (CCl_4_), the GPx levels fall from 4850 to 1000 Ul/L; meanwhile, mice treated with CCl_4_ plus extract show an increase in GPx levels, although only the group treated with 250 mg/kg extract reaches significant recovery levels above 2000 Ul/L (Figure 6). Thus, the hybrid extract may be listed as an antioxidant agent, similar to other plant extracts such as *Emblica officinalis* or *Terminalia arjuna*, for which an increase in GPx levels in erythrocytes exposed to prooxidant agents was also reported [37,38]. For GPx to convert hydrogen peroxide into water and participate in the neutralization of lipid peroxides, its antioxidant role is unquestionable [39], so the elevation of the GPx concentration in mouse serum by the extract clearly indicates that the chayote hybrid possesses antioxidant activity in vivo.

In cases where the antioxidant system cannot compensate for the excess of prooxidant agents, there is a destabilization of the cells and therefore of the tissue of which they are constituents, and this may result in chronic noncommunicable diseases (NCDs) such as type 2 diabetes mellitus, cerebrovascular accidents, heart attack, cancer or chronic liver diseases, among others [1]. This impacts epidemiology and public health since these diseases represent the top five causes of death in the world despite conventional (allopathic) therapeutic alternatives to treat each of these diseases [3].

The extract of *Sechium* H387 07 fruits contains 16 polyphenols, versus 10 in an edible *Sechium* [5], and all are present in greater concentrations; additionally, four of the most abundant in the extract enter the bloodstream and allow it to exert a systemic effect, as revealed by the reduction in serum proinflammatory cytokines of treated mice and the prevention of the fall in GPx levels in erythrocytes from mice treated with carbon tetrachloride, a powerful prooxidant agent. These observations suggest that this extract has the potential for clinical use in diseases where the antioxidant system cannot compensate for the excess of prooxidant agents, such as chronic noncommunicable diseases.

The therapeutic potential of the hybrid extract becomes relevant if it is considered that recently, only 10 phenolic compounds out of 20 possible in the extract of edible *Sechium edule* var. *nigrum spinosum* fruits were shown to be associated with relevant biomedical effects, such as a reduction in glucose levels, without damage to the liver, spleen or thymus, or white blood cells in bone marrow and even an increase in lymphocytes, monocytes and granulocytes in peripheral blood in mice, in addition to an increase in the mitotic index in bone marrow cells, which reveals the absence of toxicity (5). On the other hand, upon administration of this same compound in humans, there is a tendency to lower glucose and increase GPx, resulting in a significant reduction in TNFα, but without an effect on IL-10 levels (13). Therefore, the fact that the hybrid contains 20 phenolic compounds out of 20 possibly associated with a clear increase in GPx levels (Figure 6) and a reduction in TNFα, but an increase in the anti-inflammatory cytokine IL-10 (Table 2), as reported in this work, coupled with the fact that the hybrid extract also increases the mitotic index in bone marrow cells without the induction of apoptosis (16), encourage us to consider that the extract of the hybrid with a higher phytochemical content has greater potential for clinical use in diseases associated with oxidative inflammation stress than its edible relative and thus addresses the WHO’s task to find alternative treatments for these pathologies with high mortality rates [3,40].

## 3. Materials and Methods

### 3.1. Material and Chemicals

Plant material: Fruits of the *Sechium* H387 07 hybrid (Reg. 1344 SNICS) at horticultural maturity, specifically 18 ± 2 days after anthesis, genetically developed by the Interdisciplinary Research Group of *Sechium edule* in Mexico, A.C. (GISeM), were studied. These fruits were obtained from a mesophyll mountain forest with an altitude of 1340 m, 80 CNVV 85% ambient humidity, well-drained, slightly acidic soil (pH 6.5) that was rich in humus [20], a mean annual precipitation of 2000 mm, a photoperiod up to 12 h with respect to flower bud initiation, and a temperature of 20 °C. Immediately after collection, the fruits were cleaned, cut into small pieces, dried at 45 °C to a moisture level of 10%, and ground to a standardized particle size of 2 mm (mesh No. 10). The plant material was then stored under aseptic conditions at room temperature until use.

Chemicals: Standards for the flavonoids rutin, phloridzin, myricetin, quercetin, naringenin, phloretin, apigenin and galangin or standards for the phenolic acids (caffeic, gallic, chlorogenic, vanillic, *p*-hydroxybenzoic, *p*-coumaric, ferulic and syringic acids, trifluoroacetic acid (TFA), acetonitrile (ACN) and methanol for HPLC (≥99.0%)) were from Sigma (St. Louis, MO, USA). Sodium hypochlorite solution (4–5%), gallic acid, carbon tetrachloride (CCl_4_), phosphate-buffered saline (PBS), and ethanol (96%) were purchased from Sigma-Aldrich (St. Louis, MO, USA). 1,2-dimyristoyl-sn-glycero-3-phosphocholine (DMPC) and 1,2-dimyristoyl-sn-glycero-3-phosphoethanolamine (DMPE) were obtained from Avanti Polar Lipids Inc. (Alabaster, AL, USA). Drabkin’s reagent and the glutathione peroxidase (GPx) RANSEL kit were obtained from Randox Laboratories Ltd. (Crumlin, UK). Sodium pentobarbital (PISA, CDMX, Mexico), methanol (99.8%), Folin–Ciocâlteu reagent (Merck, Darmstadt, Germany), and TNFα, IL-6, IFNγ, IL-1β, and IL-10 antibody anti-mouse inflammatory cytokine kits (BD Biosciences, San Jose, CA, USA) were also used.

### 3.2. Extraction

The fruits of the H387 07 hybrid were cut into small pieces, dried at 45 °C to 10% moisture and ground to a standard particle size of 2 mm. Next, 1013 g of plant material was extracted in a batch with methanol (99.8%, ACS, Merck, Darmstadt, Germany) for 48 h at room temperature (20 ± 2 °C), and the resultant alcoholic extract was filtered with paper 12 times, repeating the addition of the solvent until the macerated product was not colored. Then, the solvent was evaporated at 50 °C under reduced pressure (Buchi Rotavapor R-114, Flawil, Switzerland) until a crude extract, without organic solvent, was obtained [41]. A stock solution was made with 71.2 mg of crude extract solubilized with 300 µL of ethanol (96%) and 700 µL of phosphate-buffered saline (PBS; Sigma, St. Louis, MO, USA), and corresponding dilutions were made.

### 3.3. Identification of the Total Phenolic Content (TPC)

The total polyphenol content was determined by molecular spectrophotometry at 765 nm using the Folin–Ciocâlteu reagent (Merck, Darmstadt, Germany) method [42]. Total flavonoids were determined by the Chang et al. [43] method. Gallic acid was used as the standard for the calibration curve, and the total polyphenol content was expressed as gallic acid equivalents (GAE) [44].

### 3.4. Identification of Compounds by High-Performance Liquid Chromatography (HPLC)

The extract was analyzed using a Hewlett Packard 1100 series high-performance liquid chromatograph (HPLC) with an autosampler (Agilent Technologies, 1200 Series Mod. G1329A, Santa Clara, CA, USA) and a diode array detector.

For the flavonoids, the analyses were performed on a Hypersil ODS (125 × 40 mm) Hewlett Packard Column with a gradient of (A) H_2_O at pH 2.5 with trifluoroacetic acid (TFA) and (B) acetonitrile (ACN) and the following parameters: flow, 1 mL/min; temperature, 30 °C, injection volume: 20 µL; λ1: 254 nm, λ2: 316 nm, and analysis time 25 min. The standards used were rutin, phloridzin, myricetin, quercetin, naringenin, phloretin, apigenin and galangin.

For phenolic acid, a Nucleosil column (125 × 4.0 mm) from Macherey-Nagel was used with a gradient of (A) H_2_O at pH 2.5 with trifluoroacetic acid (TFA) and (B) acetonitrile (ACN). The other experimental parameters included the following: flow, 1 mL/min; temperature, 30 °C; injection volume: 20 µL; λ: 280 nm; and analysis time 25 min. Caffeic, gallic, chlorogenic, vanillic, *p*-hydroxybenzoic, *p*-coumaric, ferulic and syringic acids were used as the standards.

External standards (Sigma, Co.) were used to identify and quantify each of the flavonoids or phenolic acids mentioned above; solutions of pure compounds were prepared at concentrations of 0.08, 0.16, 0.32, 0.64 and 1.28 mg/mL in HPLC-grade methanol. Using readings from each series of standard solutions for areas of peak absorption and flavonoid or phenolic acid concentration, linear regression equations were obtained to calculate the content of the compounds in the samples.

### 3.5. In Vitro Antioxidant Activity

DPPH test: The 2,2-diphenyl-L-picrylhydrazyl (DPPH) free radical scavenging assay was employed to evaluate antioxidant activities as previously described [45]. Aliquots (500 µL) of solutions containing gallic acid (17 µg/mL) as an antioxidant positive control or extract at concentrations of 0.5, 0.75, 1, 1.25 and 1.5 mg/mL were transferred to test tubes and mixed with 500 µL of methanol and 2 mL of 0.1 mM DPPH solution in methanol. The reaction mixtures were incubated for 30 min at room temperature in the dark, and absorbances were then measured at 517 nm. The control comprised 0.1 mM DPPH solution without extract, and the blank was pure methanol. All measurements were performed in triplicate. The percentage of DPPH inhibition was calculated according to Equation (1), in which A0 is the absorbance of 0.1 mM DPPH solution and A1 is the absorbance of 0.1 mM DPPH solution containing the sample.
DPPH inhibition (%) = [(A0 − A1)/A0] × 100(1)

The concentration of extract required to scavenge 50% of DPPH radicals (IC_50_ value) was established from dose–response data by linear regression.

Protection of model membrane structure: The capacity of hybrid *Sechium* H387 07 to interact with DMPC and DMPE multilayers was evaluated using X-ray diffraction. Two milligrams of each phospholipid were placed in Eppendorf tubes and then supplemented with 150 μL of (a) distilled water (control) and (b) aqueous solutions of extract (0.7–36.1 µg). The protective capacity of the extract was evaluated by preincubating each phospholipid with the extract for 10 min and thereafter with 10 ng of HClO. All samples were incubated for 30 min in a shaking bath at 30 °C for DMPC and 60 °C for DMPE, transferred into 1.5-mm diameter special glass capillaries (WJM-Glas, Berlin, Germany) and finally centrifuged at 2500 rpm for 15 min (MRC, Holon, Israel). A Bruker Kristalloflex 760 (Karlsruhe, Germany) generator with Ni-filtered CuKα radiation was used for the X-ray diffraction experiments. The relative reflection intensities were collected in an MBraun PSD-50M linear position sensitive detector system (Garching, Germany). The experiments were carried out at 18 ± 1 °C, which is lower than the main phase transition temperature of DMPC (24.3 °C) and DMPE (50.2 °C) [46]; higher temperatures would have induced transitions to more fluid phases, hampering the detection of structural changes. Each experiment was repeated at least twice.

### 3.6. In Vivo Antioxidant Activity

Identification of phytochemicals in serum: Female mice of the CD-1 strain (*n* of 5 mice per dose) were treated intraperitoneally with the current administration route to carry out studies on antioxidant effects [47] with the extract of the *Sechium* H387 07 hybrid, with a single inoculation for each dose, under the following scheme: 0 (PBS as vehicle), 8, 125, and 250 mg/kg, less than 1900 mg/kg established as the LD_50_ [48], and mice after treatment were sacrificed at 1, 3, 6, 12, and 24 h. To obtain serum, the mice were anesthetized with 6.3 mg/kg sodium pentobarbital (PISA, CDMX, Mexico) administered intraperitoneally. Once sedated, their blood was extracted by the axillary plexus, recovering blood in Vacutainer tubes for serum (Becton Dickinson (BD), San Jose, CA, USA). After collection, the serum was incubated for 10 min at room temperature and centrifuged for 10 min at 2000 rpm to separate the serum from the globular package. The serum was collected in tubes and frozen in liquid nitrogen for lyophilization. The lyophilization was carried out in a B-585 glass oven (Buchi, Germany) to complete dryness. The obtained lyophilizate was weighed and diluted in HPLC-grade absolute methanol and centrifuged for 10 min at 2000 rpm. The supernatant was recovered, filtered at 0.22 µm (Millipore, Ireland) and placed in HPLC vials. HPLC analysis was performed according to the protocol described above.

Detection of inflammatory cytokines: The mice were treated under the aforementioned conditions, and groups of 10 CD-1 mice were treated intraperitoneally for 30 days every 48 h with doses of 0 (PBS), 8, 250, 500 and 1000 mg/kg extract of the *Sechium* H387 07 hybrid. At the end of the treatment, the mice were anesthetized with pentobarbital and the blood of the axillary plexus was collected in serum vacutainers (BD, San Jose, CA, USA) and centrifuged at 2000 rpm for 10 min to separate the serum from the globular package. The serum was collected, and the analysis of inflammatory cytokines was performed using the Cytometric Bead Array (CBA) technique with a mouse inflammatory cytokine kit (BD, San Jose, CA, USA). Briefly, the serum was incubated with synthetic beads (loaded with antibodies against TNFα, IFNγ, IL-10, and IL-6) and phycoerythrin as a reporter for 3 h. At the end of the incubation, they were washed with buffer solution and centrifuged at 2500 rpm for 10 min, and the supernatant was removed to resuspend the pellet in 100 µL of buffer solution. The samples were read in a flow cytometer (FACSAria II, BD, San Jose, CA, USA). The concentrations of each cytokine were obtained using FCAParray 3.0 software (BD, San Jose, CA, USA).

Detection of glutathione peroxidase activity: Following the treatment protocol approved by the ethics committee for animal experiments of the Faculty of Higher Studies Zaragoza, National Autonomous University of Mexico (UNAM) (FESZ/DEPI/CI/216/14), two days before slaughter, oxidative stress was induced in the mice by administering 1 dose of 10% CCl_4_ in olive oil intraperitoneally every 24 h. At the end of this period, the mice were anesthetized to obtain blood in a Vacutainer with heparin (BD, San Jose, CA, USA), which was agitated and separated for analysis by the RANSEL kit (Randox Laboratories Ltd., Crumlin, UK), and to evaluate the activity of glutathione peroxidase (GPx). The GPx concentration was evaluated by the decrease in absorption at 340 nm due to the oxidation of dihydronicotinamide-adenine dinucleotide phosphate (NADPH) to nicotinamide adenine dinucleotide phosphate (NADP+). Briefly, 0.05 mL of heparinized whole blood was placed in 1 mL of diluent solution (provided by Randox) and incubated for 5 min. Then, 1 mL of Drabkin’s reagent was added at double concentration. Samples were analyzed in the next 20 min for the test, which included 0.02 mL of diluted sample, 1 mL of working reagent (glutathione 4 mmol/L, glutathione reductase ≥ 0.5 U/L and NADPH 0.34 mmol/L) and 0.04 mL of cumene (cumene hydroperoxide 0.18 mmol/L). Samples were mixed, and the initial absorbance was read at the end of 1 min. The timing was started simultaneously to read again after 1 and 2 min. The kinetics of this reaction were read at 340 nm on a Shimadzu UV-1600 Series brand spectrophotometer (Knauer, Japan).

### 3.7. Statistical Analysis

All analyses were performed using Microsoft Excel (Redmond, WA, USA) and IBM SPSS Statistics (version 20; Corporation, Chicago, IL, USA). All values are expressed as the means ± SD. The statistical analysis was performed through an analysis of variance (ANOVA) and Tukey’s test, with statistical significance at *p* ≤ 0.05.

## 4. Conclusions

The extract of the hybrid of *Sechium* H387 07, an intentionally generated plant, contains 16 polyphenols used as standards versus the 10 detected in edible *Sechium*, and all polyphenols were found at higher concentrations in H387 07. Additionally, four of the most abundant polyphenols in the extract enter the bloodstream of the treated mice, which allows the extract to exert a systemic reach. In vitro studies indicate that the extract has antioxidant activity. In vivo, the extract induces a reduction in proinflammatory cytokines in the serum of treated mice and prevents the decrease in GPx levels in erythrocytes from mice treated with carbon tetrachloride, a powerful prooxidant agent. Our findings suggest that this extract of the *Sechium* hybrid has anti-inflammatory and antioxidant effects in vitro and in vivo in a mouse model. For this reason, the results support the possibility of exploring the clinical effect in humans.

## Figures and Tables

**Figure 1 molecules-25-04637-f001:**
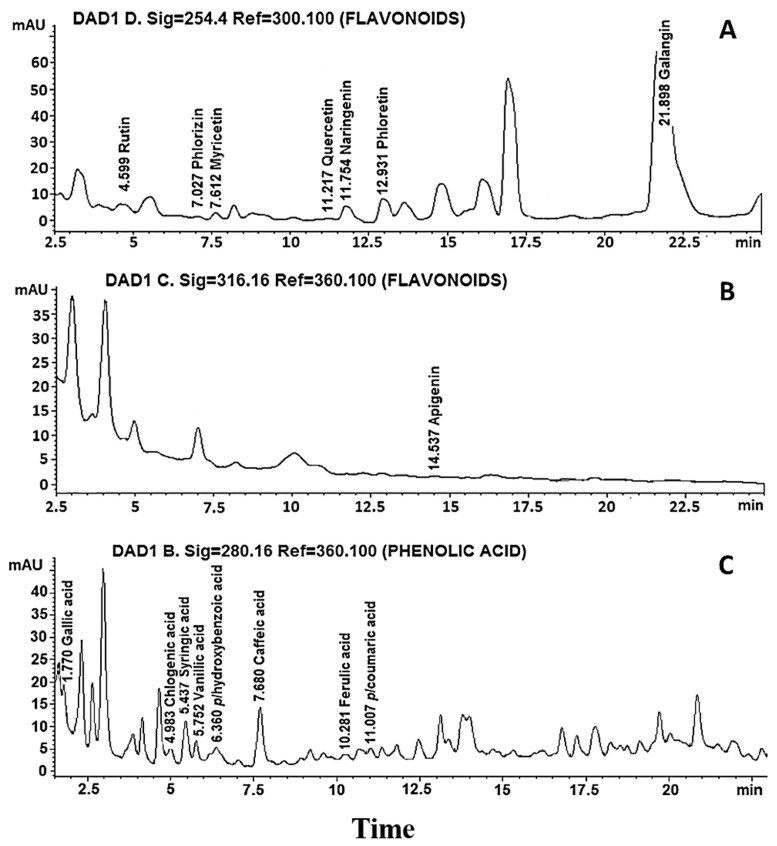
Identification of compounds present in the H387 07 hybrid fruit extract by HPLC. (**A**) Chromatograms of the flavonoids at 254 nm; (**B**) presence of apigenin at 316 nm and unidentified compounds; and (**C**) phenolic acids at 280 nm.

**Figure 2 molecules-25-04637-f002:**
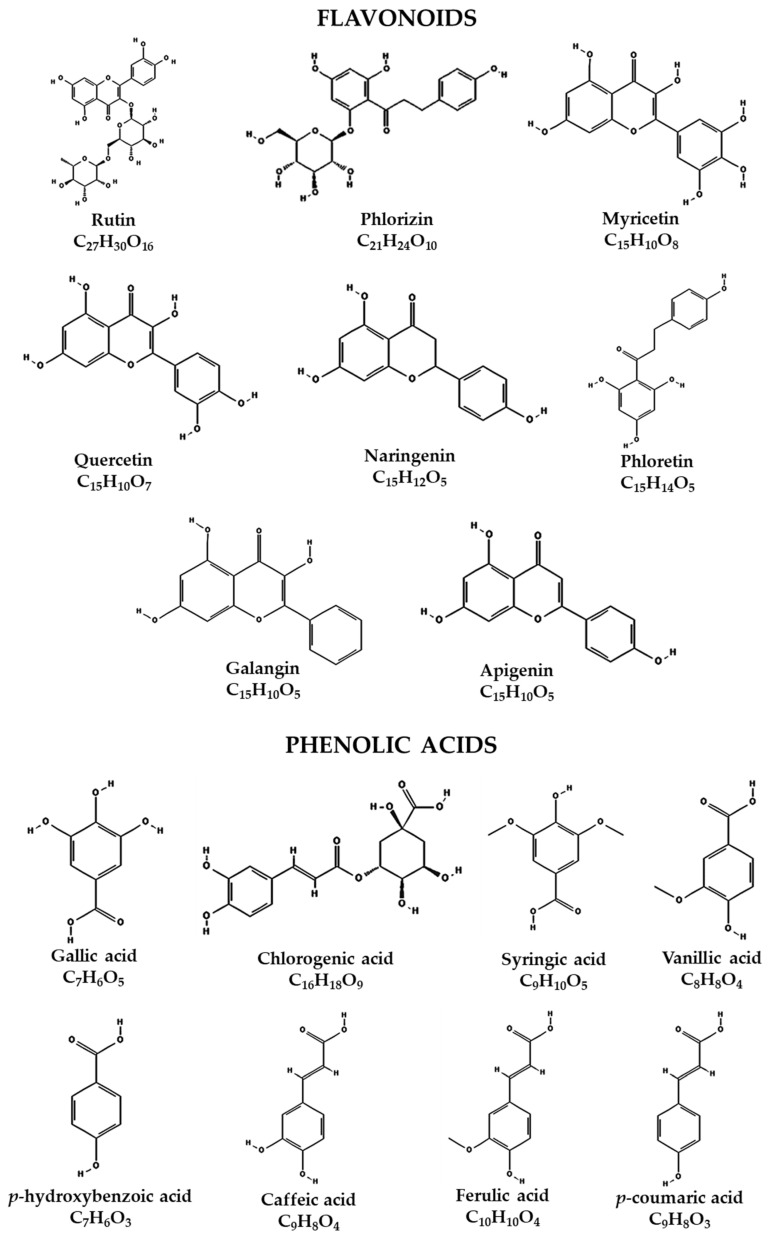
Chemical structure of flavonoids and phenolic acids identified in the extract of fruits of hybrid H387 07.

**Figure 3 molecules-25-04637-f003:**
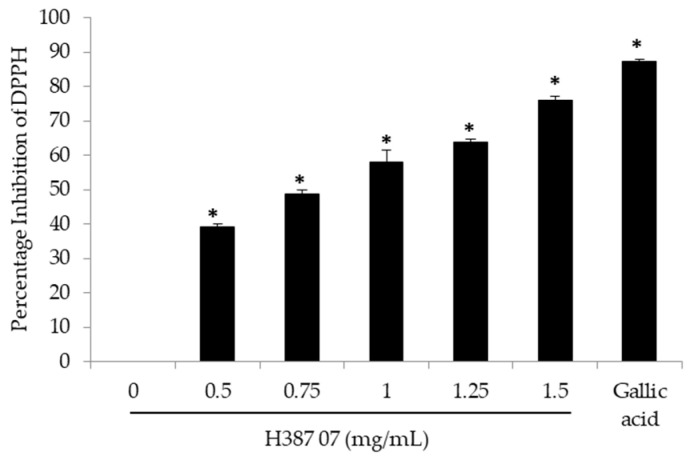
Inhibition of 2,2-diphenyl-L-picrylhydrazyl (DPPH) activity by extract H387 07. The average of three independent experiments ± standard deviation, * Significant difference (Tukey *p* < 0.05).

**Figure 4 molecules-25-04637-f004:**
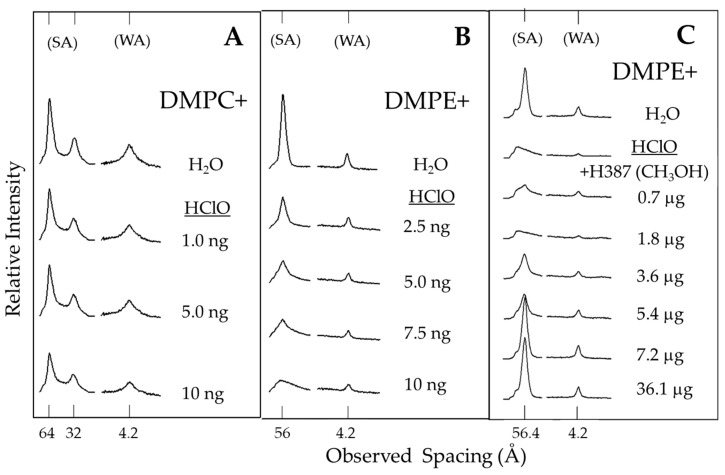
X-ray diffraction patterns of (**A**) dimyristoylphosphatidylcholine (DMPC) and (**B**) dimyristoylphosphatidylethanolamine (DMPE) bilayers immersed in water and hypochlorous acid (HClO); (**C**) DMPE in water in the presence of or absence of HClO 10 ng/mL, and with increasing concentrations of the extract of H387 07 hybrid; small-angle (SA) and wide-angle (WA) reflections.

**Figure 5 molecules-25-04637-f005:**
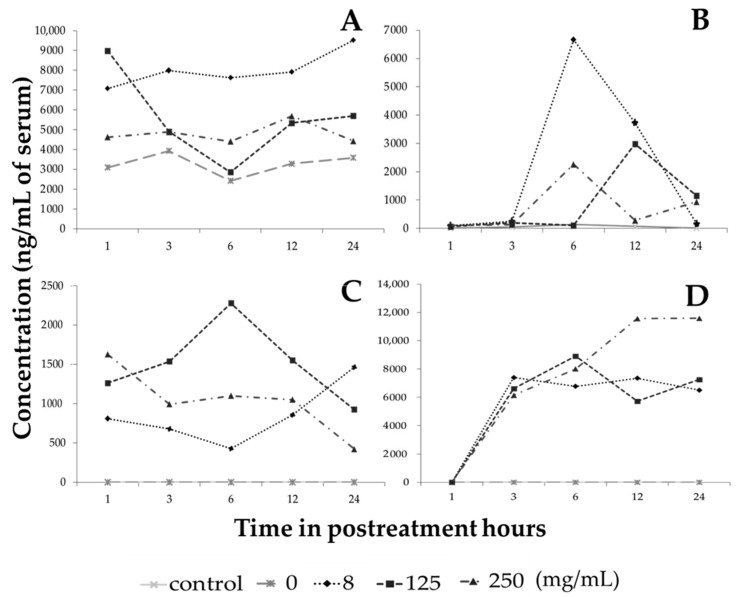
Concentration of secondary metabolites present in serum from mice treated with the extract of the *Sechium* H387 07 hybrid. (**A**) Galangin, (**B**) phloretin, (**C**) naringenin, and (**D**) chlorogenic acid.

**Figure 6 molecules-25-04637-f006:**
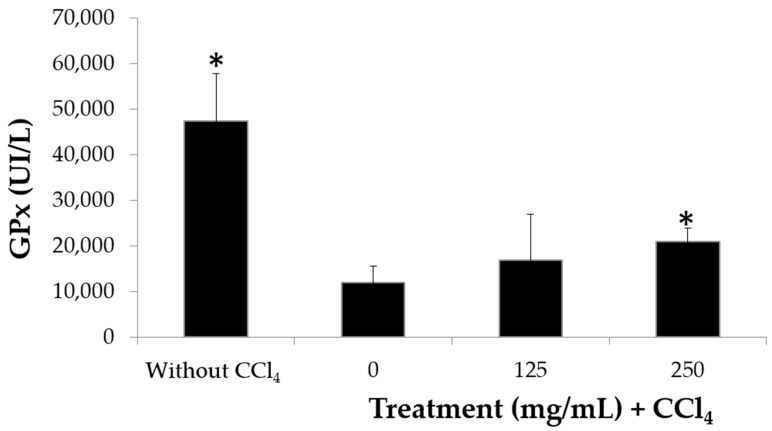
Concentration of glutathione peroxidase glutathione peroxidase (GPx) in blood lysates of mice treated or not treated with the H387 07 extract for a month with or without induction of oxidation by carbon tetrachloride (CCl_4_). *n* = 10 mice per treatment ± standard deviation. * Significant difference (Tukey *p* < 0.05).

**Table 1 molecules-25-04637-t001:** Concentration of flavonoids and phenolic acids in the extract of fruits of hybrid H387 07 obtained from HPLC.

H387 07 Hybrid Extract			
Flavonoids		Phenolic Acids	
(mg/g of Extract)			
Rutin	1.273	Gallic acid	0.056
Phlorizin	0.0168	Chlorogenic acid	4.224
Myricetin	0.889	Syringic acid	0.016
Quercetin	0.005	Vanillic acid	0.087
Naringenin	3.304	*p*-hydroxybenzoic acid	0.084
Phloretin	4.616	Caffeic acid	0.187
Galangin	21.940	Ferulic acid	0.064
Apigenin	0.362	*p*-coumaric acid	0.029

**Table 2 molecules-25-04637-t002:** Concentrations of serum inflammatory cytokines from mice treated intraperitoneally with 8, 250, 500 and 1000 mg/kg extract of the *Sechium* H387 07 hybrid.

Cytokine (pg/mL)	Without Treatment	H387 07 Hybrid Extract (mg/kg)				
		0	8	250	500	1000
TNFα	255.7 ± 37	257.9 ± 26	144.8 ± 42 *	200.4 ± 9 *	143.1 ± 23 *	92.46 ± 25 *
INFγ	69.7 ± 6	56.6 ± 3	49.6 ± 7	27.5 ± 8 *	19.07 ± 6 *	26.9 ± 3 *
IL-6	590.0 ± 56	512.5 ± 60	125.8 ± 47 *	378.7 ± 20 *	231 ± 41 *	237 ± 27 *
IL-10	342.6 ± 75	276.9 ± 82	1070 ± 375 *	2831 ± 168 *	3870 ± 480 *	4156 ± 200 *

*n* = 10 mice per treatment ± standard deviation, * ANOVA, significant difference (Tukey *p* < 0.05).

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
