# Peer review of "Phytochemical Analysis and Antioxidant and Anti-Inflammatory Capacity of the Extracts of Fruits of the Sechium Hybrid"

_molecules, 2020, doi:10.3390/molecules25204637_

Round 1
Reviewer 1 Report
In the manuscript entitled “Phytochemical analysis and antioxidant and anti-inflammatory capacity of the extract of fruits of the Sechium hybrid.” the authors evaluate the chemical composition, the antioxidant and anti-inflammatory capacity of the extract of fruits of the Sechium hybrid using in vivo and in vitro models. The work is overall well done, carefully thought, and performed and the manuscript is well written and easy to read and follow. All experimental methods are well explained. The methods used are consistent with the literature and corroborate the objectives. The results presented are significant, robust and their presentation and interpretation are supported by other data present in the literature. The conclusions are supported by the results presented.
Other Specific comments:
The chemical characterizations have been well described; however, important chemical compounds with hight concentration no are identified (a compound with TR= 16 and 17.5 min – Figure A / compounds with TR= 2.5 and 5 min – Figure B / compounds with TR= 1.8 and 2.7 min – Figure C). I suggest using the HPLC/MS/MS to verify.
I suggest inserting a new paragraph showing a comparison of the chemical profile of the new Sechium H387 07 hybrid and Sechium edule variety.
The results presented in this manuscript seem to confirm the existence in other studies as paper, which were used as references, however, a relevant contribution of this work would be present if these results were directly compared with the anti-inflammatory and antioxidant actions present in the literature making a direct correlation with chemical composition showing the difference between Sechium H387 07 hybrid and Sechium edule variety. This important point is can be used in the presentation and discussion of the results as the perspective of future use.
The quality of English writing throughout the manuscript needs minor adjusting. Native or professional English writer assistance may be required
Author Response
Reviewer 1
Specific comments:
The chemical characterizations have been well described; however, important chemical compounds with hight concentration no are identified (a compound with TR= 16 and 17.5 min – Figure A / compounds with TR= 2.5 and 5 min – Figure B / compounds with TR= 1.8 and 2.7 min – Figure C). I suggest using the HPLC/MS/MS to verify.
Answer: Thanks for the suggestion. Indeed, there are, in the indicated running times, peaks that suggest a high concentration of phytochemicals not identified in the extracts of the hybrid but absent in its parent. To emphasize this particularity, it was supplemented in results and discussion, page 3, paragraph 2, line 108… In fact, the retention time from 15 to 17.5 min at 254.4 nm (Figure 1 A) or 2.5 to 5 min at 316.16 nm (Figure 1B) for flavonoid compounds and 1.8 to 2.7 min (Figure 1C) for phenolic acids, the hybrid contains high concentrations of unidentified compounds that are not present in its relative (5), ….
I suggest inserting a new paragraph showing a comparison of the chemical profile of the new Sechium H387 07 hybrid and Sechium edule variety.
Answer: Thanks for the suggestion. The same is attended by pointing out phytochemicals present in the hybrid but not in its parent. We added line 117…The hybrid contains rutin, myricetin, quercetin and galangin, or syringic acid and ferulic acid, which are all absent in its parent, on the other hand, ….
The results presented in this manuscript seem to confirm the existence in other studies as paper, which were used as references, however, a relevant contribution of this work would be present if these results were directly compared with the anti-inflammatory and antioxidant actions present in the literature making a direct correlation with chemical composition showing the difference between Sechium H387 07 hybrid and Sechium edule variety. This important point is can be used in the presentation and discussion of the results as the perspective of future use.
Answer: Thanks for the suggestion. In keeping with the recommendation and to highlight the anti-inflammatory and antioxidant properties of the relative and to point out that the hybrid with the highest content of phytochemicals may have a greater biomedical impact. The last paragraph of results and discussion line 305 is restructured.…... The therapeutic potential of the hybrid extract becomes relevant if it is considered that recently, only 10 phenolic compounds out of 20 possible in the extract of edible Sechium edule var. nigrum spinosum fruits were shown to be associated with relevant biomedical effects, such as a reduction in glucose levels, without damage to the liver, spleen or thymus, or white cells in bone marrow and even an increase in lymphocytes, monocytes and granulocytes in peripheral blood in mice, in addition to an increase in the mitotic index in bone marrow cells, which reveals the absence of toxicity (5). On the other hand, upon administration of this same compound in humans, there is a tendency to lower glucose and increase GPx, resulting in a significant reduction in TNFα but without an effect on IL-10 levels (13). Therefore, that the hybrid contains 20 phenolic compounds out of 20 possibly associated with a clear increase in GPx levels (Figure 6) and reduction of TNFα but increase of the anti-inflammatory cytokine IL-10 (Table 2), as reported in this work, coupled with the fact that the hybrid extract also increases the mitotic index in bone marrow cells without induction of apoptosis (16), encourage us to consider that the extract of the hybrid, with higher phytochemical content, has greater potential for clinical use in diseases associated with oxidative-inflammation stress than its edible relative and thus addresses the WHO’s task to find alternative treatments for these pathologies with high mortality rates [3,40].
The quality of English writing throughout the manuscript needs minor adjusting. Native or professional English writer assistance may be required.
Thank you. The manuscript was edited by professional English writer assistance (American Journal Experts).
Reviewer 2 Report
The submitted manuscript molecules-929822 falls within the scope of the Journal Journal and investigate the phytochemical analysis and antioxidant and anti-inflammatory capacity of the extract of fruits of the Sechium hybrid.
My suggestions have been carefully considered and Authors have made the corresponding changes. The manuscript can be accepted in present form.
Authors are only invited to translate in English the following sentence (lines: 107-108): "Estos datos muestran mayor contenido y riqueza de fitoquimicos en el hibrido H387 en comparación con uno de sus progenitors."
Author Response
Reviewer 2
Authors are only invited to translate in English the following sentence (lines: 107-108): "Estos datos muestran mayor contenido y riqueza de fitoquimicos en el hibrido H387 en comparación con uno de sus progenitors."
Answer: Tank you. We translate the sentence. Line 125… These data show higher content and richness of phytochemicals in hybrid H387 compared to one of its progenitors.
Reviewer 3 Report
The manuscript “Phytochemical analysis and Antioxidant and anti-inflammatory capacity of the extract of fruits of the Sechium hybrid.” needs to be carefully revised prior its publication in Molecules. For example, the quality of the English is poor and need to be carefully checked by a native speaker.
Major concerns:
1) English needs a careful correction by a native speaker . Manuscript shows a lot of grammatical mistakes, typos, lack of concordances between subject and verbs, between pronouns/articles and nouns, lack of articles, bad use of pronouns (for example, use of “it is” when it should be “its”. And so on. Example of recurrent typo: please change “explorer” by “exploring” in introduction and conclusions (and anywhere else if used in other sections of the manuscript).
2) Mass spectrometry should be performed to try to elucidate more flavonoids.
3) Please add a subsection, in material and methods, for the chemical compounds used, indicating provider, city and country. Information is disseminated along all section, and incomplete.
Minor concerns
1) Please revise presentation and appearance. There are different interlining space sizes along the manuscript. Please homogenize in accordance with Journal guidelines.
2) Introduction may need a few additional references.
3) Please translate to English the lines 107 and 108.
4) Table 1 could include also the mass of unidentified flavonoids, to show how much percentage (in mass) of them have been appropriately identified.
Author Response
Reviewer 3
The manuscript “Phytochemical analysis and Antioxidant and anti-inflammatory capacity of the extract of fruits of the Sechium hybrid.” needs to be carefully revised prior its publication in Molecules. For example, the quality of the English is poor and need to be carefully checked by a native speaker.
Major concerns:
1) English needs a careful correction by a native speaker . Manuscript shows a lot of grammatical mistakes, typos, lack of concordances between subject and verbs, between pronouns/articles and nouns, lack of articles, bad use of pronouns (for example, use of “it is” when it should be “its”. And so on. Example of recurrent typo: please change “explorer” by “exploring” in introduction and conclusions (and anywhere else if used in other sections of the manuscript).
Thank you. The manuscript was edited by professional English writer assistance (American Journal Experts).
2) Mass spectrometry should be performed to try to elucidate more flavonoids.
Answer: Thanks for the observation. Please take into consideration that lacking HPLC / MS equipment and this data must be analyzed in future studies. We indicate that this pending line 112……so it would be interesting in the future to use a more sensitive method for identification, such as HPLC/mass spectrometry.
3) Please add a subsection, in material and methods, for the chemical compounds used, indicating provider, city and country. Information is disseminated along all section, and incomplete.
Answer: Thanks for the observation. We add a subsection, line 336
Chemicals. Standards for flavonoids of rutin, phloridzin, myricetin, quercetin, naringenin, phloretin, apigenin and galangin or standards for phenolic acid of caffeic, gallic, chlorogenic, vanillic, p-hydroxybenzoic, p-coumaric, ferulic and syringic acids, trifluoroacetic acid (TFA), acetonitrile (ACN) and methanol for HPLC (≥99.0%) were from Sigma (St. Louis, MO, USA). Sodium hypochlorite solution (4-5%), gallic acid, carbon tetrachloride (CCl4), phosphate buffered saline (PBS), and ethanol (96%) were purchased from Sigma-Aldrich (St. Louis, MO, USA). 1,2-dimyristoyl-sn-glycero-3-phosphocholine (DMPC) and 1,2-dimyristoyl-sn-glycero-3-phosphoethanolamine (DMPE) were obtained from Avanti Polar Lipids Inc. (Alabaster, AL, USA). Drabkin’s reagent and glutathione peroxidase (GPx) RANSEL kit were obtained from Randox Laboratories Ltd. (Crumlin, UK). Sodium pentobarbital (PISA, CDMX, Mexico), methanol (99.8%) and Folin-Ciocalteu reagent (Merck, Darmstadt, Germany), and TNFα, IL-6, IFN-α, IL-1β, and IL-10 antibody anti-mouse inflammatory cytokine kits (BD Biosciences, San Jose, CA) were also used.
Minor concerns
1) Please revise presentation and appearance. There are different interlining space sizes along the manuscript. Please homogenize in accordance with Journal guidelines.
Thank you. The manuscript was review.
2) Introduction may need a few additional references.
Thank you. Please take into consideration that in the previous review, we were invited to include no more than 45 citations, but it could only be lowered to 48, so we would be grateful if you consider that we have already exceeded the limit of appointments.
3) Please translate to English the lines 107 and 108.
Answer: Tank you. We translate the sentence. Line 125… These data show higher content and richness of phytochemicals in hybrid H387 compared to one of its progenitors.
4) Table 1 could include also the mass of unidentified flavonoids, to show how much percentage (in mass) of them have been appropriately identified.
Thanks for the observation. Please take into consideration that lacking HPLC / MS equipment, this information is not available at the moment. We know that this data must be analyzed in future studies.
Round 2
Reviewer 3 Report
The manuscript "Phytochemical analysis and Antioxidant and anti-inflammatory capacity of the extract of fruits of the Sechium hybrid.” has improved from the next round and I accept the answers to the concerns from the Authors in the non-fixed concerns. Thus, I recommend its acceptance.
Just only I would like to remember Authors, for next works, the importance of having HPLC-MS in these type of studies.
This manuscript is a resubmission of an earlier submission. The following is a list of the peer review reports and author responses from that submission.
Round 1
Reviewer 1 Report
In the manuscript entitled “Phytochemical analysis and antioxidant and anti-inflammatory capacity of the extract of fruits of the Sechium hybrid” the authors determine the antioxidant and anti-inflammatory capacity of the extract of fruits of the Sechium hybrid. The work is overall well done, carefully thought and performed and the manuscript is well written and easy to read and follow. All experimental methods are well explained. Other Specific comments:
Why used the intraperitoneal route in the activity in vivo?
In vivo test, which LD50? If not, how can we ensure that the doses tested do not have the potential for harm?
Explain why used the CCl4 as agent inflammatory? This compound normally is used in the hepatic damage assay.
In the HPLC analysis - The authors need to determine the limit of detection (LOD) and limit of quantification (LOQ) used. The validation of the method and the quantification must be performed following ICH procedures (ICH, 2012). I would like underline that this aspect is very important to answer to the requirements of the journal.
What is the criterion used to determine if the dose used in in vivo assay? The dose is selected by LD50? Explain.
The results presented in this manuscript seem to confirm the existing in other studies as paper to used as an example to indicate as cite, however, this important point is not used for presentation and discussion of the results as the perspective of future treatment.
Please correlate the value of IC50 for all antioxidant assay with the content of phenols and flavonoids of crude extract. (Suggest use Pearson correlations)
There is a vast literature demonstrating the relation of anti-inflammatory activity. It is not clear what the real contribution of this work is since there are other articles that demonstrate the anti-inflammatory activity? In this sense, it will be the interest of the work to show this action and how the present work collaborates to clarify this relationship. Please see
The quality of English writing throughout the manuscript is inferior, it is not necessary to list all errors and unprofessional expressions. Native or professional English writer assistance may be required
Reviewer 2 Report
The submitted manuscript molecules-783665 falls within the scope of the Journal and investigate the phytochemical analysis and antioxidant and anti-inflammatory capacity of the extract of fruits of the Sechium hybrid. The main drawbacks of this manuscript are related to the lack of data on phenolic fraction and the poor information about compound identification, leading to consider this manuscript not adequate to publish in Molecules. However, the manuscript deserves other ameliorations.
TITLE
The paper title is well stated, it is informative and concise.
ABSTRACT
The following sentence: “A Sechium hybrid, show one thousand times greater antineoplastic activity than edible species,….” is not a result of this paper and should be modified or deleted.
RESULTS AND DISCUSSION
The results obtained in this study are very interesting, but there is a lack of information about the extract of “H387 07 hybrid”, because, many peaks of the chromatograms of the flavonoids and phenolic acids were not identified (See Figure 1).
Lines 266-271: Delete the following sentences because they are repeated two times: “The therapeutic potential of the hybrid extract becomes relevant if it is considered that recently, it was published that the concentrate of Sechium edule variety nigrun spinosum fruits induces antioxidant effects in patients with metabolic syndrome [55], a fact that stimulates consideration that the extract of the hybrid, with a higher phytochemical content, has the potential for clinical use in diseases associated with oxidative stress and thereby addresses the WHO’s task to find treatment alternatives to these pathologies with high mortality rates [3,56]. (See Lines 260-265)
MATERIAL AND METHODS
Material and research methods are presented appropriately and clearly. Unfortunately, the methodology and devices used by authors in this manuscript did not allow to have a deep characterization of phenolic profile of this Sechium hybrid. It's a shame, because the chromatograms presented includes several important peaks. Why the authors did not use the MS and NMR to perform a complete identification?
Lines 285-286: Delete the following sentence because it is repeated two times: “Fruits of the H387 07 hybrid were cut into small pieces, dried at 45°C to 10% moisture and ground to a standardized particle size of 2 mm.” (See Lines 280-281)
It is recommended to authors to explain the test employed in the statistical analysis section. In fact, in the paper the Authors cited “Dunnett’s comparison test” (see Line 391), while in Tables and Figures the Authors cited “Tukey” (See: Line 130 in Figure 2 - Line 222 in Table 2 - Line 244 in Figure 5). Please, clarify!
CONCLUSION
The authors stated that: “The extract of the hybrid of Sechium H387 07, a plant intentionally generated, contains 16 polyphenols versus 10 for an edible Sechium, and all are present at greater concentrations.”, but they do not take in account many unidentified compounds showed in the chromatograms of Figure 1. This sentence is not correct.
LITERATURE
Too many references are cited in this paper; 35-45 references are reasonable for such a paper (correct the references, delete the oldest references and avoid excessive auto-citation). Furthermore, some references reported in the list of references diverge from the Journal style guidelines and require some attention.
Reviewer 3 Report
The manuscript "Phytochemical analysis and Antioxidant and anti-inflammatory capacity of the extract of fruits of the Sechium hybrid." reports interesting findings and could be published in Molecules after revising the English and the concerns given below:
English:
In general English is readable, but a careful revision, preferably by a native speaker, should be done to correct minor things as punctuation, usage of articles a/an/the, typos and so on. Please check also the grammar, as mistakes can be found. Example at conclusions:
"The extract of the hybrid of Sechium H387 07, a plant intentionally generated, contains 16 polyphenols versus 10 for an edible Sechium, and all are present at greater concentrations. Additionally, four of the most abundant in the extract..." Better "versus the 10 detected in an edible Sechium, and all polyphenols were found in higher concentrations at H387 07. Additionally, four of the most abundant polyphenols in the extract...".
Major concerns
1) All the names of species should be in italics (Sechium edule). Please correct in all the manuscript. ame applies for the genus, when cited (Sechium).
2) Adding a scheme with the chemical structures of the phenolic compounds (for example, rutin, phlorizidine, mirecetin, quercetin, naringenin, phloretin, galangin, apigenin; and any other relevant studied in the manuscript) will help the readers to associate the chemical structures. It will help also to fill the big blank space in page 3, improving the quality of the presentation, and thus making the article more attractive.
3) A positive control (for example, ascorbic acid) needs to be added to Figure 2.
Minor concerns
1) Introduction can have any mentioning of the free radicals
2) Presentation of Table 1 needs to be improved, to have less interlining between rows.
3) Besides, caption of tables and figures do not follow journal format guidelines. Please revise and correct.
4) Please make clearer the units in Table 2, as the units of TNFalfa, INFgamma and interleukines are not given and seeing the table is like it is mg/kg, when this unit is of the extract.